# Robot path planning based on tabu particle swarm optimization integrating cauchy mutation

Lishu Qin
*The college of mechanical engineering*
*Dalian University*
Dalian, China
qinlishu@dlu.edu.cn

Zhentao Fan
*The college of mechanical engineering*
*Dalian University*
Dalian, China
fanzhentao47@gmail.com

*Abstract*—In view of the shortcomings of the traditional Particle Swarm Optimization (PSO) in robot path planning, such as long path planning time, slow convergence speed, insufficient search ability in the middle and late stages, and easy to fall into local optimum. Inspired by the Tabu Search (TS) algorithm and Cauchy Mutation Perturbation, this paper proposes a Tabu Particle Swarm Optimization (PSOTS) based on Cauchy Mutation. Firstly, the beta distribution random number strategy is used to adaptively adjust the inertia weight to improve the search accuracy of the algorithm in the middle and late stages. Secondly, the Cauchy mutation perturbation is introduced in the particle swarm optimization (PSO) stage to update the position of particles, so as to reduce the possibility of particles falling into the local optimum. Finally, in the middle and late stages of the search, the roulette method was used to select some particles and adopt the Tabu Search (TS) algorithm, so as to enhance the local search ability of the particles in the middle and late stages. Through the test of CEC benchmark function, compared with the traditional particle swarm optimization (PSO) algorithm, it is proved that it has excellent performance with fewer iterations and running time, and compared with the test results of specific obstacle environments, PSOTS can effectively generate the optimal path with high smoothness and shorter length, and improve the convergence speed and stability of PSO, which proves its superiority in solving robot path planning problems.

*Keywords—particle swarm optimization, adaptive, cauchy mutation, tabu search*

## I. INTRODUCTION

The path planning of mobile robots is a very hot research topic at present, and it is widely used in the fields of deep-sea mining vehicle navigation [1], robotic arm operation [2], laser cutting [3], unmanned aerial vehicles (UAVs) [4], and unmanned vehicles [5]. Path planning refers to the robot's ability to find a collision-free optimal or suboptimal path from the current position to the target location under the constraints of mechanical conditions and complex environments [6]. Considering the type and number of robots, environmental complexity, static or dynamic obstacles, etc. It is a challenging task to plan a path with a short path and a high degree of smoothness without colliding with obstacles and other robots. To solve such problems, scholars have established environmental models to simulate the actual engineering environment, relied on path search algorithms to find feasible paths that meet the constraints, and obtained the optimal path through fitness function screening [7].

Path planning algorithms can be broadly divided into two categories, namely traditional methods and intelligent optimization algorithms. Traditional path planning algorithms include A-star algorithm [8], Dijkstra algorithm [9], D-star algorithm [10], and Fast Search Random Tree Algorithm (RRT). [11] and Artificial Potential Fields (APF) [12]. Among them, A-star, Dijkstra, and D-star are path planning algorithms based on graph theory. For this type of approach, a suitable mesh model needs to be built. However, in the grid method, the mesh size is susceptible to environmental constraints. In addition, for complex environments, graph-theoretic algorithms are computationally expensive and inefficient [13]. The RRT algorithm can quickly generate barrier-free paths, but it cannot guarantee the optimal solution of paths, and the path smoothness is insufficient. The advantage of the APF algorithm is that the planning speed is fast, but when the attraction and repulsion are equal, it will fall into a situation where the target is unreachable, making it impossible for the robot to move towards the target position. Traditional methods are based on mathematical models and can be applied to any environment, but they tend to fall into local minima.

The path planning problem of mobile robots is an NP-Hard problem. To solve this problem, many scholars have adopted heuristic algorithms combined with intelligent optimization, including gray wolf algorithm (GWO) [14], ant colony algorithm(ACO)[15],particle swarm optimization (PSO) [16], differential evolution algorithm (DE) [17], reinforcement learning (RL) [18], genetic algorithm (GA) [19], etc. For most of the path planning literature, only single-objective optimization is carried out on the path length factor, and the path smoothness and safety factors are not considered, resulting in the problems of unsatisfactory planning path, insufficient energy, and long search time [20].

For the research of particle swarm optimization, many scholars have innovated in two aspects. On the one hand, it is an improvement of the particle swarm optimization itself, including the parameter adaptive strategy, the redefinition of the velocity function, the perturbation strategy, etc., the purpose of which is to make the particles jump out of the local optimum, improve the search ability and convergence speed. Focus on the early and middle stages of particle search for improvements; On the other hand, it is the combination of particle swarm optimization and other algorithms, which can be combined with genetic algorithm (GA), differential evolution (DE), reinforcement learning (RL), etc. using the advantages of other algorithms to make up for the lack of search ability of particle swarm algorithm.

In order to solve the above problems, this paper proposes a Tabu Particle Swarm Optimization algorithm (PSOTS) Integrating Cauchy Mutation, which is applied to the two-dimensional environment. The algorithm improves the adaptive strategy of inertia weight $\omega$, and adopts the Cauchy mutation strategy in the particle update stage to reduce the probability of particles falling into the local optimum. In order to improve the mid-to-late search ability of the algorithm, the Tabu Search (TS) algorithm was applied to the mid-to-late stage of particle search. In addition, the corresponding fitness functions are designed for the two factors of path length and angle change.

The major contributions of this paper are summarized as follows:

- An adaptive strategy is adopted for the inertia weights to improve the global search ability of particles.

- The Cauchy Mutation is applied to the particle position update stage to reduce the probability of particles falling into the local optimum.

- The Tabu Search Algorithm (TS) and Particle Swarm Optimization (PSO) were combined to improve the convergence speed and mid-to-late search ability of the traditional particle swarm algorithm.

- The simulation tool is used to verify the effectiveness of the algorithm in different environments, and compared with the traditional particle swarm optimization to prove its superiority.

The remaining sections of this paper are organized as follows. Section 2 describes the related work of particle swarm optimization. Section 3 is the global modeling of the obstacle environment and the formulation of the fitness function. Section 4 briefly introduces the traditional particle swarm optimization and tabu search algorithm. Section 5 presents the improved algorithm in detail. Section 6 is the simulation experiment part. Finally, section 7 deals with conclusion and future work.

## II. RELATE WORK

### A. Improvements to the Algorithm Itself

Li et al. [21] proposed a Fermat-based Grouped Particle Swarm Optimization (FP-GPSO) algorithm to simultaneously determine the aerial launch position and optimize the generated multi-segment path for the path planning problem of composite UAV paths. Zhao et al. [22] proposed an unmanned vehicle path planning method based on Adaptive Particle Swarm Optimization (APSO). Firstly, the map simplification strategy (MSS) is adopted to simplify the search space, and then the search of particles is coordinated by three adaptive factors and the Levy flight strategy, and the safety check strategy and dynamic obstacle avoidance strategy are proposed to ensure the safety of the global path. Tao et al. [23] proposed a two-population PSO algorithm (BPPSO) with a stochastic perturbation strategy, which divides particles into two subpopulations. The first population enhances the global search capability by taking into account the mass of the particles and the optimal solution of the randomly selected particles when updating the speed. The second population uses a linear cognitive coefficient adjustment strategy to enhance the local search. Xu et al. [24] proposed a nonlinear dissipative particle swarm algorithm. The algorithm dissipates particles in a nonlinear increment way, which avoids a large amount of unnecessary dissipation at the beginning of the iteration, and invests more effort in dissipation at the end of the iteration, which improves the operation efficiency and global search ability of the algorithm. Wang et al. [25] proposed an improved immune particle swarm algorithm. Among them, the adaptive information dynamic adjustment strategy is introduced to dynamically adjust the main link index, which improves the global searchability and convergence of particles, and is conducive to the robot to quickly identify the optimal path.

### B. Combination between Algorithms

Lin et al. [26] combined the cultural algorithm with the particle swarm optimization and introduced a probabilistic method based on improved metropolitan rules to update the inertia weights, which solved the path planning problem of multiple AGVs. Zhao et al. [27] proposed a new multi-objective Cauchy Mutant Cat Population Optimization (MOCMCSO) and Artificial Potential Field Method (APFM) to solve the multi-objective optimization problem of the shortest global path length and the smallest total rotation angle change. Wu et al. [28] proposed a new path planning algorithm combining ant colony optimization and particle swarm optimization. First, the simulated annealing algorithm is combined with the particle swarm for global search. The ant colony algorithm is used for local searches at a later stage. Mohammed et al. [29] combined Informed Rapid Exploration Random Tree (RRT*) and particle swarm optimization (PSO) algorithms. The informed RRT* algorithm can quickly construct the optimal path, which, combined with the particle swarm algorithm, increases the speed of algorithm convergence. Huang et al. [30] proposed an APSO algorithm combining A* and PSO to calculate the optimal path, and used the redundant point removal strategy to preliminarily optimize the path planned by the A* algorithm to obtain the key node set. After that, the improved PSO optimization key node set is used to obtain the global path.

## III. ENVIRONMENT MODELING AND FITNESS FUNCTION

### A. Environment Modeling

In this paper, the grid method is used to describe the working environment of a mobile robot. As shown in Figure 1, the white grid represents the feasible node, the purple grid represents the obstacle node, the green grid represents the starting point, and the yellow grid represents the end point. The raster method can represent the complex working environment of the robot in a concise two-dimensional vector. Therefore, this paper creates a 20×20 mesh model with random obstacle nodes. All grids are arranged in left-to-right, bottom-to-top order, and each grid has a unique ordinal number. The starting node is the 0th grid, and the number 0 represents the starting point. The terminating node is the 399th raster, and the number 399 indicates the end point.

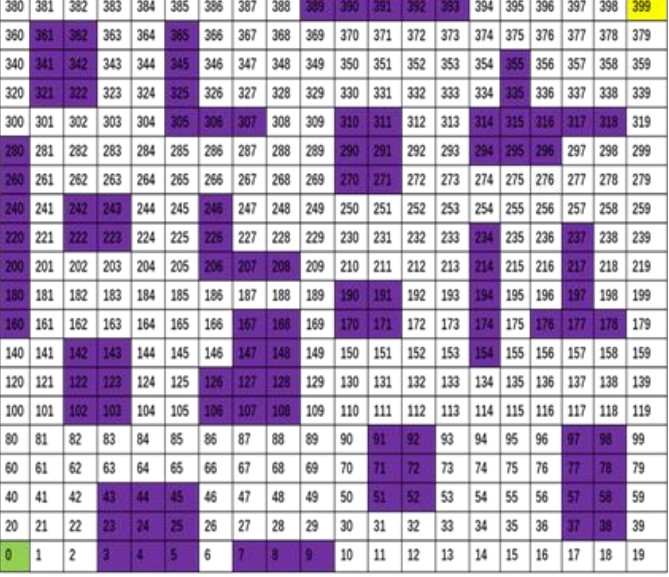

Fig.1.Grid Environment Diagram of Mobile Robot.

### B. Fitness Function

In the path planning problem, to find the optimal path, it is necessary to meet the constraints such as path length, safe obstacle avoidance, and turning angle limitations. The fitness function combines these constraints to transform the path planning problem into an optimization problem. The fitness function is used as the basis for evaluating the quality of particles, which determines the speed and direction of particle update iteration. With reference to Ref. [31], the fitness function is designed in the following form:

$$f(x) = \frac{1}{fitness} = \frac{1}{\alpha * f_{length} + \beta * f_{smooth}} \tag{1}$$

where $fitness$ is the fitness function, $f(x)$ is the objective function, $f_{length}$ is the fitness function of the path length, $f_{smooth}$ is the fitness function of the path smoothness, and $\alpha$, $\beta$ are the weights of the two fitness functions. It should be noted that the fitness function is the reciprocal of the objective function, so the

smaller the objective function, the higher the particle fitness. $f_{length}$ and $f_{smooth}$ are defined as follows:

$$\begin{cases} f_{length} = \dfrac{\gamma_1}{length} \\ f_{smooth} = \dfrac{\gamma_2}{smooth} \end{cases} \tag{2}$$

where $length$ is the path length, $smooth$ is the path smoothness, and $\gamma_1$, $\gamma_2$ are the accuracy factors. By adjusting $\gamma_1$, $\gamma_2$, it can be guaranteed that $f_{length}$, $f_{smooth}$ in the same order of magnitude.

$$length = \sum_{i=1}^{n-1} \sqrt{\left(x_{i+1} - x_i\right)^2 + \left(y_{i+1} - y_i\right)^2} \tag{3}$$

Its smoothness can be calculated using Eq. (4):

$$smooth = \sum_{i=2}^{n-1} smooth_i \tag{4}$$

where $smooth_i$ indicates the degree of smoothness between adjacent segments, and this value is designed based on the angle $\varphi_i$ between two segments. The following equation (5) is given:

$$smooth_i = \begin{cases} 0 & \varphi_i = 0 \\ 5 & 0 < \varphi_i < \dfrac{\pi}{2} \\ 30 & \varphi_i = \dfrac{\pi}{2} \\ 50 & \varphi_i > \dfrac{\pi}{2} \end{cases} \tag{5}$$

The angles between adjacent segments are obtained by the inverse cosine function, as follows equation (6)-(8):

$$\begin{cases} a_1 = \left(x_{i-1} - x_i\right)^2 + \left(y_{i-1} - y_i\right)^2 \\ b_1 = \left(x_{i+1} - x_i\right)^2 + \left(y_{i+1} - y_i\right)^2 \\ c_1 = \left(x_{i-1} - x_{i+1}\right)^2 + \left(y_{i-1} - y_{i+1}\right)^2 \\ a = \sqrt{a_1} \\ b = \sqrt{b_1} \\ c = \sqrt{c_1} \end{cases} \tag{6}$$

$$d_i = \frac{a_1 + b_1 + c_1}{2 * a * b} \tag{7}$$

$$\varphi_i = \arccos\left(d_i\right) \tag{8}$$

## IV. TRADITIONAL ALGORITHMS

### A. Particle Swarm Optimization (PSO)

The particle swarm optimization (PSO) was proposed by James Kennedy and Russell Eberhart [1]. In the PSO algorithm, the initial particle population is randomly generated, and the local and global best particles are selected based on the fitness function values of each particle. At each iteration, the PSO algorithm updates the particle position based on Equation (9) and (10). After the position of each particle is updated, a fitness function is calculated based on the results obtained, the global optimal particle is determined, and the cycle continues until the termination condition is true.

$$V_i^d = \omega v_i^d + c_1 r_1 (pbest_i^d - x_i^d) + c_2 r_2 (gbest_i^d - x_i^d) \quad (9)$$

$$X_i^d = x_i^d + V_i^d \quad (10)$$

Where $c_1$, $c_2$ are the acceleration coefficients; $\omega$ is the inertia weight, which is adaptive in this paper, as shown in Equation (11); $i$ indicates the current number of iterations; $d$ represents dimensions; $pbest$ represents the local optimal position; $gbest$ represents the global optimal position.

### B. Tabu Search Algorithm (TS)

The Tabu Search Algorithm (TS) simulates the human mind, and its core idea is to prohibit the repetition of previous actions. Its main advantages include two points:

1)  The addition of the taboo table and the "amnesty criterion" makes the algorithm accept some inferior solutions in the search process, so as to jump out of the trap of local optimality.

2)  The new solution is not randomly generated from the neighborhood of the current solution, but is the best solution that is not taboo.

## V. IMPROVED ALGORITHM

### A. Adaptive Inertia Weight $\omega$

For the adaptive adjustment of inertia weight $\omega$, exponential function $e^{-\frac{t}{tmax}}$ and beta distribution random numbers are introduced. The improved inertia weight expression is as follows:

$$\omega = \omega_{min} + (\omega_{max} - \omega_{min}) * e^{(-t/tmax)} + \alpha * betarnd(p,q) \quad (11)$$

Where $t$ is the current number of iterations; $t_{max}$ is the maximum number of iterations; $\omega_{max}$ is the initial inertia weight, and the value is 0.9; $\omega_{min}$ is the inertia weight when the maximum number of iteration is reached, and the value is 0.4; $\alpha$ is the inertia adjustment factor, and the value is 0.1; $betarnd(p,q)$ is a beta distribution, where $p = 2$, $q = 5$.

Where the inertia weight $\omega$ decreases nonlinearly with the increase of the number of iterations. The addition of the random number of beta distribution makes the value distribution of inertia weight more reasonable, and improves the global search ability and later search accuracy in the early stage.

### B. Cauchy Mutation Update Populations

The classical PSO algorithm does not fully take into account the information of other suboptimal solution particles when updating the population position, so the population diversity cannot be guaranteed. In this paper, the Cauchy mutation operator is introduced on the basis of the classical particle swarm algorithm. In the stage of updating the position of particles, the particles in the population undergo the mutation process under the influence of the Cauchy mutation operator. $x_i$ represents the first generation of particles, the new individual $x_i^*$ produced by Cauchy mutation. The expression is:

$$x_i^* = x_i + \sigma * tan(\pi * (rand - 0.5)) \quad (12)$$

Where $\sigma$ is a random number of the standard normal distribution, and its probability density function is:

$$f(x^*) = \frac{1}{\sqrt{2\pi}} e^{-\frac{x^2}{2}} \quad (-\infty < x < +\infty) \quad (13)$$

The variation value for the current dimension is set to standard deviation $\sigma * tan(\pi * (rand - 0.5))$, where $rand$ is a random number between 0 and 1, which conforms to a uniform distribution.

Cauchy mutation can produce a large random number space, make the particle distribution more uniform, expand the search range of particles, avoid falling into the local optimal solution, and improve the global convergence ability.

Figure 2 shows the schematic diagram of Cauchy mutation, where the yellow particles represent the local optimal particles ($pbest$) and the green particles represent the ordinary search particles.

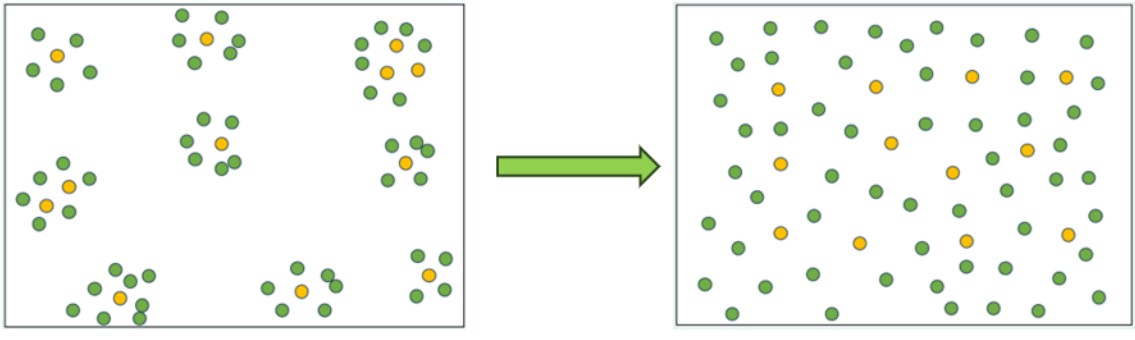

(a) Randomly Distributed Initialized Populations      (b) Cauchy Mutation Update Populations

Fig.2. Schematic Diagram of Cauchy Mutation

*C. Tabu Particle Swarm Algorithm Integrating Cauchy Mutation（PSOTS）*

This section describes the steps and flowchart for the improved algorithm. Figure 3 is a flowchart of the improved algorithm, and the steps of the improved algorithm are as follows:

**Step1:** Initialize parameters of PSO algorithm and TS algorithm.

**Step2:** Randomly generate initial population, set velocity and position of each particle.

**Step3:** Update velocity and position of each particle according to equations (9) and (10).

**Step4:** All particles perform particle swarm algorithm.

**Step5:** Determine whether *pbest* and *gbest* are updated. If not, go to the next step, otherwise go to step 3.

**Step6:** Update the population by cauchy mutation.

**Step7:** Roulette pick up particles to compose population 2 and the remainder is population 1.

**Step8:** Population 1 performs particle swarm search.

**Step9:** Population 2 performs tabu search.

**Step10:** Compare results of particle swarm search and tabu search. Output better result.

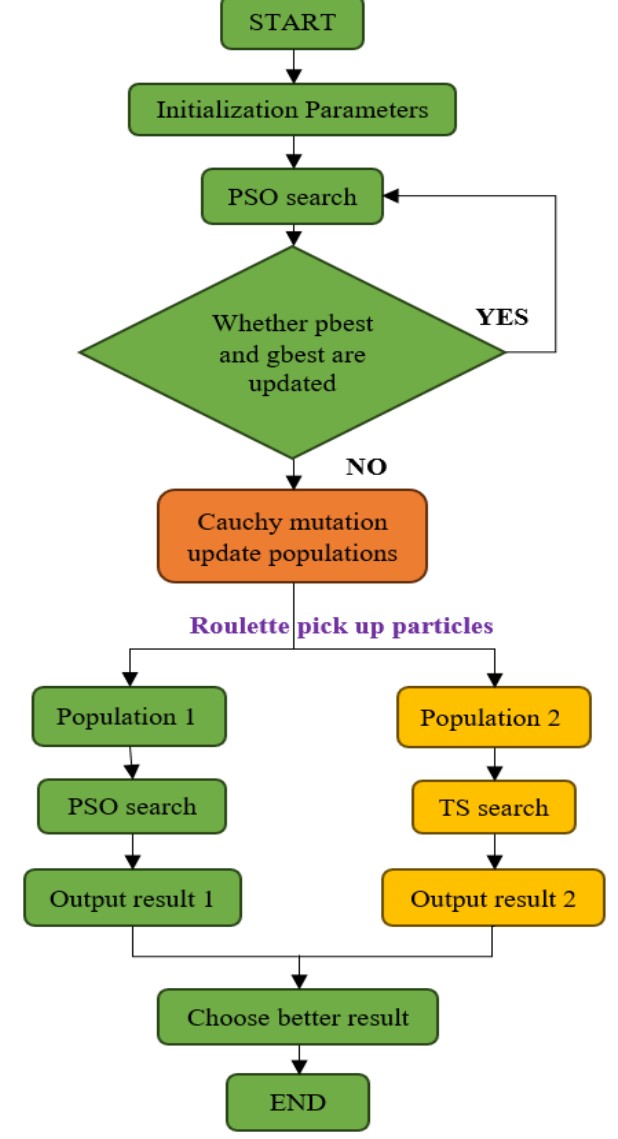

Fig. 3. Flowchart of the Improved Algorithm

## VI. SIMULATION EXPERIMENTS

### A. Parameter Settings

TABLE I shows the parameters of the improved algorithm.

### B. Experiment 1: CEC Function Test Experiment

In order to verify the optimization effect of the improved algorithm, two test functions are used to test the algorithm. Where the function $F_1(x)$ is a multimodal function and the function $F_2(x)$ is a unimodal function. The name, formula, search space, and theoretical minimum of the test function are given in TABLE II. TABLE III shows the parameters of the comparison algorithms.

Each function experiment was performed 50 times, and the results obtained were averaged. In terms of the effectiveness of the improved algorithm (PSOTS), compared with the ordinary particle swarm optimization (PSO), the test results are shown in TABLE IV, and the convergence curve of the test function is shown in Figure 4.

TABLE I. PARAMETERS OF THE IMPROVED ALGORITHM

| Parameter | Value |
|---|---|
| Particle Population | 50 |
| Particle Dimension | 2 |
| Maximum Iteration | 100 |
| $\alpha$ | 1 |
| $\beta$ | 7 |
| $\gamma_1$ | 1 |
| $\gamma_2$ | 1 |
| $c_1$ | 1.2 |
| $c_2$ | 1.2 |
| $\omega_{min}$ | 0.4 |
| $\omega_{max}$ | 0.9 |

TABLE II. TEST FUNCTIONS

| Number | Function | Equation | Search Range | $f_{min}$ |
|---|---|---|---|---|
| $F_1(x)$ | Schwefel's problem | $\sum_{i=1}^{n} -x_i \sin\left(\sqrt{|x_i|}\right)$ | $[-500,500]$ | $-12569.5$ |
| $F_2(x)$ | Step Function | $\sum_{i=1}^{n} (\lfloor x_i + 0.5 \rfloor)^2$ | $[-100,100]$ | 0 |

TABLE III. PARAMETERS OF THE COMPARISON ALGORITHMS

| Algorithm | Parameter |
|---|---|
| PSO | Iteration=300, Population=150, $\omega$=0.8, $c_1$=1.2, $c_2$=1.2 |
| PSOTS | Iteration=300, Population=150 |

TABLE IV. FUNCTION TEST RESULTS

| Function | PSO | | | PSOTS | | |
|---|---|---|---|---|---|---|
| | Global Optimizer | Standard Deviation | Convergent generation | Global Optimizer | Standard Deviation | Convergent generation |
| $F_1(x)$ | -10986.07 | 0 | 147 | **-12666.67** | 0 | **62** |
| $F_2(x)$ | 1 | 0 | 208 | 1 | 0 | **172** |

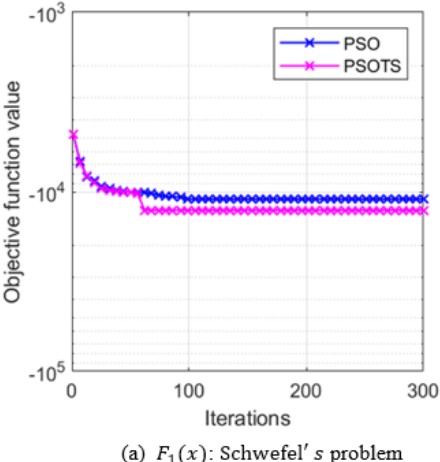

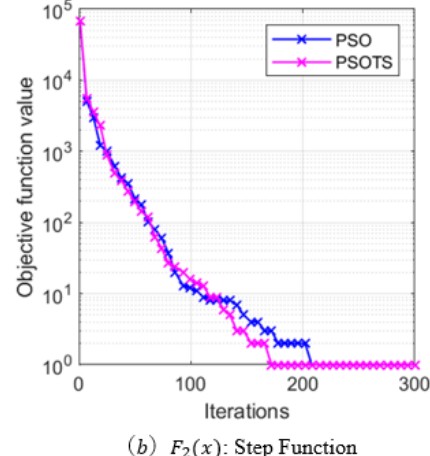

(a) $F_1(x)$: Schwefel's problem

(b) $F_2(x)$: Step Function

Fig.4. Convergence Curve

Analysis of TABLE IV shows that for test function $F_1(x)$, the global optimal solution obtained by the improved method (PSOTS) is -12666.67, while the global optimal solution obtained by the traditional method (PSO) is -10986.07, and the improved algorithm (PSOTS) is closer to the theoretical optimal solution of the function, which proves that the improved method has higher solution accuracy and is easier to jump out of the local optimum. At the same time, compared with the fastest convergence algebra, the improved method (PSOTS) requires 62 generations to complete convergence, while the traditional method (PSO) needs 147 generations to complete convergence, indicating that the convergence speed of the improved method is faster.

For test function $F_2(x)$, the global optimal solution obtained by the two algorithms is the same, but the improved algorithm (PSOTS) is fully converged in the 172nd generation, and the traditional algorithm (PSO) is fully converged in the 208th generation, indicating that the convergence speed of the improved method is faster when the solution accuracy is the same.

In summary, the improved method has higher solving accuracy, faster convergence speed, easier to get rid of local optimum, and better global convergence

### C. Experiment 2: Specific Obstacle Environment Experiment

In order to verify the performance of the proposed algorithm, a comparative experiment is designed, and eight raster obstacle environments are created. According to the number of obstacles in the environment, the area of the total environment, and the complexity of the shape, the eight environments are divided into two categories: simple environment and complex environment, and each category contains four environments.

The performance of Ordinary Particle Swarm Optimization (PSO) and Improved Particle Swarm Optimization (PSOTS) in the same obstacle environment is compared. The green line represents the path planned by the improved algorithm, and the blue line represents the path planned by the traditional particle swarm algorithm.

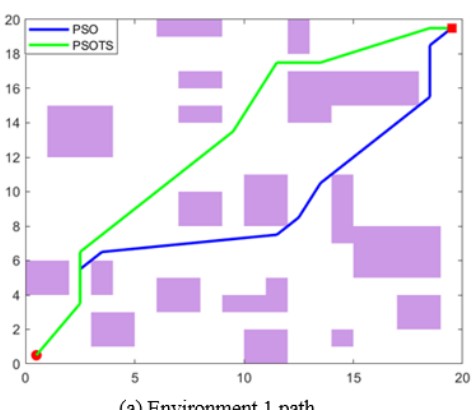

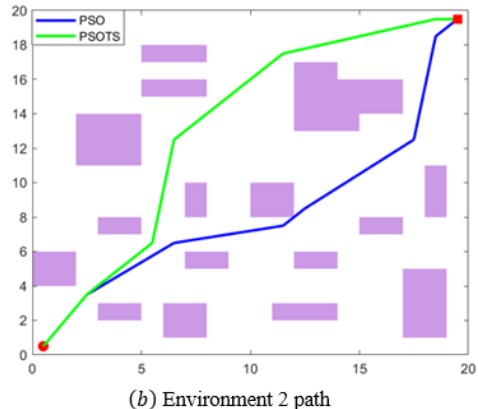

(a) Environment 1 path

(b) Environment 2 path

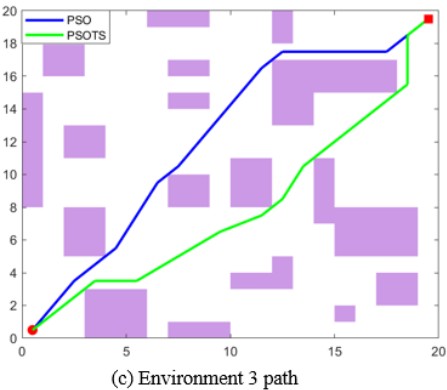

(c) Environment 3 path

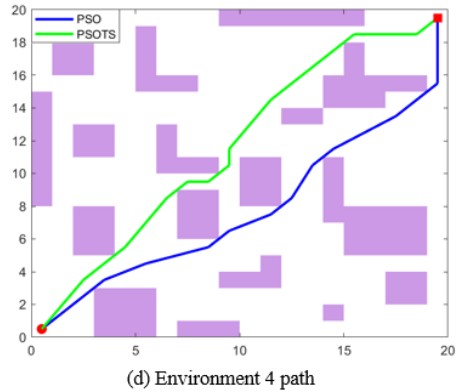

(d) Environment 4 path

Fig.5. Simple Environments Paths

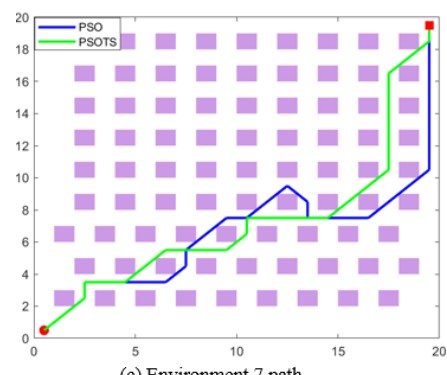

(a) Environment 5 path

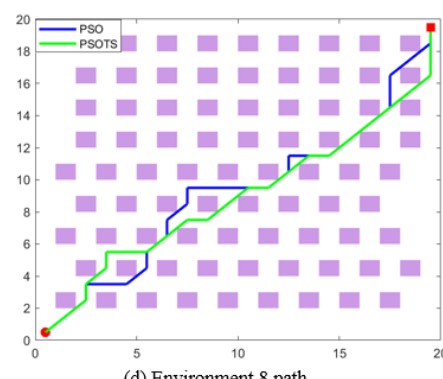

(b) Environment 6 path

(c) Environment 7 path

(d) Environment 8 path

Fig.6. Complex Environments Paths

TABLE V.     SIMPLE ENVIRONMENT PATHS RESULTS

| Algorithm | Mean Path Length | Mean Smooth | Mean Fitness | Mean $F(x)$ |
|---|---|---|---|---|
| PSO | 29.3256 | 5400 | 0.0354 | 28.2516 |
| PSOTS | 28.5045 | 4750 | 0.0366 | 27.3554 |

TABLE VI.     COMPLEX ENVIRONMENT PATHS RESULTS

| Algorithm | Mean Path Length | Mean Smooth | Mean Fitness | Mean $F(x)$ |
|---|---|---|---|---|
| PSO | 32.0725 | 18100 | 0.0316 | 31.6796 |
| PSOTS | 30.1938 | 17613 | 0.0335 | 29.8368 |

TABLE V obtains the average values of the path length, smoothness, fitness values and objective functions of the two algorithms in the four simple environments mentioned above.

Analyzing the data in TABLE V, it can be seen that on the average path length, Compared with the ordinary particle swarm algorithm, the improved algorithm is reduced by 0.8211 and reduce 2.8%, indicating that the path of the improved algorithm is shorter, and in terms of smoothness, the improved algorithm is reduced by 650 and reduce 12% compared with the ordinary particle swarm algorithm, indicating that the path of the improved algorithm is smoother, and in terms of particle fitness, the improved algorithm is 0.0012 higher than that of the ordinary particle swarm, indicating that the particle fitness of the improved algorithm is higher, and it is proved that the improved algorithm has a better effect of planning the path in a simple environment.

TABLE VI obtains the average values of path length, smoothness, fitness values and objective functions of the two algorithms in the above four complex environments.

Analyzing the data in TABLE VI, it can be seen that the average path length of the improved algorithm is 1.8787 and reduce 5.9% lower than that of the ordinary particle swarm algorithm, indicating that the path planned by the improved algorithm is shorter. In terms of smoothness, the improved algorithm is reduced by 487 and reduce 2.7% compared with the ordinary particle swarm algorithm, indicating that the path of the improved algorithm is smoother, and in terms of particle fitness, the improved algorithm is 0.0019 higher than that of the ordinary particle swarm algorithm, indicating that the particle fitness of the improved algorithm is higher, and it is proved that the improved algorithm has a good effect on path planning in complex environments.

Compare the results of the path planned by the improved algorithm in the complex environments and simple environments. It can be found that with the increase of environmental complexity, the reduction rate of the path length planned by the improved algorithm is greater, and the particle fitness is also higher. The results show that the higher the complexity of the environment, the stronger the path planning ability of the improved algorithm.

## D. Experiment 3: Iterative Comparative Experiment

In this experiment, we create an obstacle environment which the Common Particle Swarm Optimization (PSO) and the Improved Particle Swarm Optimization (PSOTS) obtain the

same path. The path is shown in Figure 7. Figure 8 shows the iterative curve of the two algorithms under environment 9.

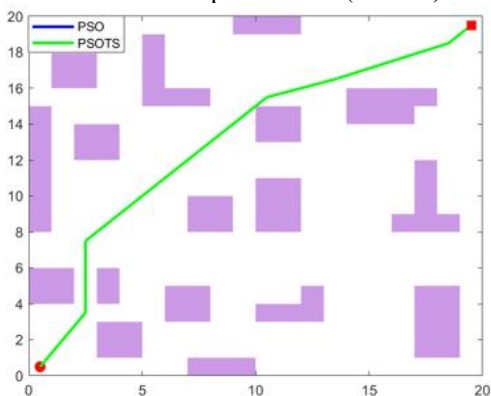

Fig.7. Environment 9 path

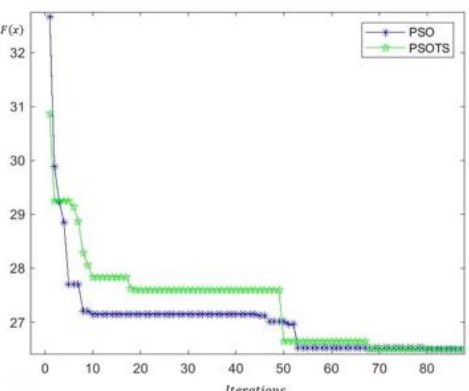

Fig.8. Iterative Curve

Comparing the iteration curve Figure 8, it can be seen that the ordinary particle swarm optimization (PSO) is fully converged in the 80th generation iteration, while the improved algorithm (PSOTS) is completely converged in the 68th generation iteration, which proves that the improved method is better than the particle swarm optimization in terms of convergence speed.

At the 0th iteration, the objective function value of the ordinary particle swarm algorithm is 32.6661, and the fitness is 0.0306. The objective function value of the improved algorithm is 30.8687, and the fitness is 0.0324. The results show that the initialized particles obtained by the improved algorithm have better effects and higher fitness values.

The longest horizontal segment of the iteration curves of the two algorithms is compared, that is, the fitness remains unchanged temporarily. The longest time of ordinary particle

swarm is from the 10th generation iteration to the 44th generation iteration, which lasts for 34 generations, while the improved algorithm lasts only 30 generations from the 19th generation to the 49th generation, which proves that the improved algorithm is superior to the ordinary particle swarm in the ability to jump out of the local optimum.

Comparing the iterative curves of the two algorithms from the 40th generation to the 60th generation, the objective function value of the ordinary particle swarm algorithm decreased from 27.1493 in the 40th generation to 26.5252 in the 60th generation, a decrease of 0.6241, while the improved algorithm decreased from 27.598 in the 40th generation to 26.6493 in the 60th generation, a decrease of 0.9487, which proves that the improved algorithm has stronger search ability in the middle and late stages.

This experiment is designed to further demonstrate the superiority of the improved algorithm in convergence speed and solution speed on the basis of test function experiments.

## VII. CONCLUSION AND FUTURE WORK

### A. Conclusion

In this paper, we propose a tabu particle swarm optimization based on cauchy mutation to improve the search ability of particle swarm optimization. Firstly, the adaptive strategy is used to balance the global and local search capabilities for the inertia weights ω , and improve the search accuracy in the middle and late stages. In addition, the cauchy mutation is introduced in the stage of particle update position, which reduces the possibility of particles falling into local optimum. Finally, the tabu search algorithm was used in the middle and late stages of the search, which enhanced the local search ability in the middle and late stages. Through the CEC test function experiment and the same path experiment, it is proved that the improved algorithm is better than the ordinary particle swarm optimization in terms of convergence speed and calculation accuracy. In different obstacle experiments, the path length and smoothness of the improved algorithm are shorter, which proves the superiority of the improved algorithm PSOTS in the path planning problem.

### B. Future Work

In more complex raster environments such as 30×30 and 40×40, etc. the adaptability of the improved algorithm is worth exploring. At the same time, the obstacle avoidance of dynamic obstacles is also one of the future issues.

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
