# OpenReview forum: "Robot path planning based on tabu particle swarm optimization integrating cauchy mutation"
_IEEE.org/ICIST/2024/Conference — IEEE ICIST 2024 Conference Submission_

### Official Review · Reviewer_e9WC · 2024-08-26
**Minor revision**

**Rating:** 6
**Confidence:** 5

**Review:**

This paper proposes a Tabu Particle Swarm Optimization algorithm (PSOTS) Integrating Cauchy Mutation, which is applied to the two-dimensional environment.
1. The abstract is too long.
2. The novelties (4 items) in introduction should be concise.
Overall, the writing and language should be improved and polished.

---

### Official Review · Reviewer_2Hj7 · 2024-08-29
**Accept after modification**

**Rating:** 6
**Confidence:** 4

**Review:**

This paper primarily propose a tabu particle swarm optimization based on cauchy mutation to improve the search ability of particle swarm optimization. Generally speaking, the overall narrative of the article is complete. However, I still have some questions as follows:

1.The initial three paragraphs of the introduction focus extensively on path planning, a topic that does not constitute the primary keyword of this paper. The extensive discussion of traditional path planning algorithms may be disproportionate. It would be beneficial to enhance the discussion regarding the current state of research on Particle Swarm Optimization (PSO) to better emphasize the unique contributions of this paper. It is recommended to revise the introduction to more clearly delineate the paper's focal points.
2.The HIGHLIGHT section of the paper is suggested to be further improved and should be compared with the current state of existing research to highlight the contribution of the paper.
The HIGHLIGHT section of the manuscript could be further refined to more effectively showcase the paper's contributions. It would be advantageous to draw comparisons with the existing body of research to better illustrate the paper's significance and advancements.
3.The RELATE WORK section, which is typically included in the introduction, appears to be misplaced. Additionally, the table in the SIMULATION EXPERIMENTS section is presented in an unappealing manner. A revision of the article's layout is recommended to enhance clarity and professionalism.
4.The purpose of Figures 5 and 6 seems to indicate differences in path planning between PSO and PSOTS. However, the demonstration of PSOTS's superiority over traditional PSO is not entirely clear. A more thorough analysis of the simulation experiments is recommended to better substantiate the advantages of PSOTS.
5.The conclusion should not merely restate the abstract but should provide a comprehensive summary of the paper's findings and implications. It is suggested that the conclusion be optimized to reflect the paper's contributions more effectively.

---

### Official Review · Reviewer_3owo · 2024-08-30
**This paper can be accepted.**

**Rating:** 7
**Confidence:** 3

**Review:**

Innovation: A Tabu particle swarm optimization algorithm (PSOTS) based on Cauchy variation is proposed. This algorithm improves the performance of PSO in robot path planning by adaptive adjustment of inertia weights, introduction of Cauchy variation perturbation and combination of tabu search algorithm, which has certain innovation.
Advantages:
1. The adaptive strategy is adopted to adjust the inertia weight ω to improve the global search ability of particles.
2. Apply Cauchy variation in the particle position update stage to reduce the probability of particles falling into local optimal.
3. Combine Tabu search algorithm (TS) and particle swarm optimization algorithm (PSO) to improve the convergence speed and late search ability of traditional particle swarm algorithm.
4. The effectiveness of the algorithm in different environments is verified by simulation experiments, and its superiority is proved by comparison with the traditional particle swarm optimization algorithm.
Cons:
1. Simple environment modeling: Although the grid method is simple and intuitive to describe the working environment of robots, the description of the complex environment may not be accurate enough, and the randomly generated obstacle nodes may not fully reflect the complexity of the actual environment, which affects the effectiveness verification of the algorithm in practical applications to a certain extent.
2. Single comparison algorithm: In the experiment, it is only compared with traditional particle swarm optimization algorithm, and lacks comparison with other advanced intelligent optimization algorithms, which is difficult to fully reflect the advanced nature of the improved algorithm in the field of path planning.

---

### Decision · Program_Chairs · 2024-09-06

Accept (Oral)